# Identification of Candidate Gene-Based Markers for Girth Growth in Rubber Trees

**DOI:** 10.3390/plants10071440

**Published:** 2021-07-14

**Authors:** Gunlayarat Bhusudsawang, Ratchanee Rattanawong, Thitaporn Phumichai, Wirulda Pootakham, Sithichoke Tangphatsornruang, Kittipat Ukoskit

**Affiliations:** 1Department of Biotechnology, Faculty of Science and Technology, Rangsit Campus, Thammasat University, Pathum Thani 12120, Thailand; g33bh@tu.ac.th; 2Nong Khai Rubber Research Center, Rubber Research Institute of Thailand, Rubber Authority of Thailand, Rattanawapi District, Nong Khai 43120, Thailand; ratchaneerattanawong@gmail.com; 3Rubber Research Institute of Thailand, Rubber Authority of Thailand, Bangkok 10900, Thailand; thitaporns@gmail.com; 4National Science and Technology Development Agency, Thailand Science Park, Khlong Luang, Pathum Thani 12120, Thailand; wirulda.poo@nstda.or.th (W.P.); sithichoke.tan@nstda.or.th (S.T.)

**Keywords:** candidate gene association mapping, intron length polymorphism, single nucleotide polymorphism, girth growth, rubber tree

## Abstract

Girth growth is an important factor in both latex and timber production of the rubber tree. In this study, we performed candidate gene association mapping for girth growth in rubber trees using intron length polymorphism markers (ILP) in identifying the candidate genes responsible for girth growth. The COBL064_1 marker developed from the candidate gene (*COBL4*) regulating cellulose deposition and oriented cell expansion in the plant cell wall showed the strongest association with girth growth across two seasons in the Amazonian population and was validated in the breeding lines. We then applied single molecule real-time (SMRT) circular consensus sequencing (CCS) to analyze a wider gene region of the *COBL4* to pinpoint the single nucleotide polymorphism (SNP) that best explains the association with the traits. A SNP in the 3’ UTR showing linkage disequilibrium with the COBL064_1 most associated with girth growth. This study showed that the cost-effective method of ILP gene-based markers can assist in identification of SNPs in the candidate gene associated with girth growth. The SNP markers identified in this study added useful markers for the improvement of girth growth in rubber tree breeding programs.

## 1. Introduction

Girth growth of the rubber tree, *Hevea brasiliensis* (Willd.ex Adr.de Juss.) Muell. Arg., is the most important indicator of the maturity of the plantation, on which the harvesting of latex is based. Wood derived from the trunk of the rubber tree, harvested during periodic replanting, has emerged as an important raw material, especially in counties where logging of the natural forest is restricted. Breeding for girth growth is time-consuming, as the rubber tree has a long juvenile period, non-synchronous flowering, low fruit setting, and a heterozygous nature. Propagation of new varieties requires a minimum of 20–25 years of experimentation. To improve the rubber productivity, a deeper knowledge of the genetic factors responsible for variation in girth growth is necessary.

Genome-wide association mapping for girth growth in Amazonian accessions grown in a suboptimal climate zone has identified candidate genes for adaptation to suboptimal climate zone [1]. However, none of the functional genes associated with secondary growth had been in the SNP set analyzed previously. The low linkage disequilibrium (LD) was also reported in the Amazonian population. Successful application of genome scanning to outcrossing species such as rubber tree will require a very high number of markers, making the candidate gene method more attractive [1,2]. The candidate gene approach has proven successful in many instances, such as in Arabidopsis [3] wheat [4,5], pea [6], potato [7], and perennial ryegrass [8]. Secondary growth has been extensively studied, and several of the genes involved have been identified. Candidate genes for cellulose and lignin biosynthesis and cell wall development, and transcription factors involved in wood development, have been reported for loblolly pine [9] and eucalyptus [10,11,12,13]. The allelic variation of these candidate genes may be used to establish associations between targeted genes with known function and girth growth in rubber tree.

Gene-based or functional markers are derived from sequence polymorphisms within genes that are directly associated with phenotypic variation. Gene-based markers accurately discriminate between traits associated with alleles of a target gene and are ideal molecular markers for marker assisted selection in breeding [14]. Although high throughput sequencing techniques are being developed nowadays, the use of variable markers present in candidate genes is still an interesting alternative for association studies. An approach known as intron length polymorphism (ILP), based on the alignment of EST cDNA sequences against known genomic sequences, has been applied to a range of crop species for genetic diversity, evolutionary and association genetic studies [15,16,17,18,19,20,21]. As ILP markers are gene-based, co-dominant, neutral, convenient, reliable and cost effective, and can play a vital role in regulation of gene expression [22], this is a promising approach to genetic studies such as candidate gene association mapping, which seeks to identify allelic variation in the genes that influence girth growth.

In the present study, we developed a set of ILP markers from candidate genes involved in secondary growth and performed candidate gene association mapping for girth growth in Amazonian accessions. Rubber tree breeding lines were used to validate the effectiveness of the ILP markers identified by association mapping. We applied targeted sequencing using Pacific Biosciences single molecule real-time (SMRT) circular consensus sequencing (CCS) technology to determine the sequence variation of the strongest associated candidate gene influencing girth growth. We then conducted candidate gene association mapping in order to exploit SNPs for downstream applications to support rubber tree breeding programs. Our study showed that the cost-effective method of ILP gene-based markers can assist in identification of SNPs in the candidate gene associated with girth growth. The SNP markers identified in this study added useful markers for the improvement of girth growth in rubber tree breeding programs.

## 2. Results

### 2.1. Development and Identification of ILP Markers

ILP variation was investigated in 26 genes from four function classes. A total of 45 ESTs (Appendix A) were aligned with whole-genome shotgun contigs of the rubber tree to predict the position and number of the introns. The 45 ESTs showed significant hits with 158 contigs of the whole-genome shotgun sequences. Of 530 predicted introns, 235 introns with sizes smaller than 800 bp were selected for primer synthesis and used in the evaluation of length polymorphism. Of the 235 ILP primer pairs examined, 228 pairs successfully amplified genomic DNA samples. We selected 115 polymorphic ILP markers (Appendix A) based on their predetermined intronic-Indel size and PIC. The average size of the amplicon was 284 bp, ranging from 150 (CesA004) to 759 bp (PAL117). The selected ILP markers yielded 470 alleles, with an average of 4.20 alleles per locus. The number of alleles ranged from 2 to 12. PIC values were between 0.06 and 0.88, with an average of 0.44. The observed heterozygosity ranged from 0.02 to 1 and the expected heterozygosity from 0.10 to 0.92, with average values of 0.59 and 0.47, respectively.

### 2.2. Population Structure and LD Analysis of ILP Markers

We further tested the 115 polymorphic ILP markers to evaluate the genetic population structure of the Amazonian accessions used in the present study. The model-based Bayesian analysis, combined with computation of the Evanno ΔK statistics (Appendix A), suggested two clusters (K = 2) with admixed accessions. Cluster 1 (green bars) comprised 13 accessions from Acre, one from Mato Grosso, and 71 from Rondonia. Cluster 2 (red bars) comprised 52 accessions from Mato Grosso and eight from Rondonia (Appendix A). The accession with unknown origin (BRAZIL_UN) was placed into Cluster 2. Twenty-four accessions had a membership probability (Q value) below 0.7 and were defined as admixtures. Of these, one originated from Acre, 11 from Mato Grosso, and 12 from Rondonia. The PCA results (Appendix A) were consistent with those from the admixture model. The accessions from Mato Grosso formed a group that was distinct from the Acre and Rondonia cluster.

The population LD was analyzed using the 115 polymorphic ILP markers. Based on *r*^2^ estimates, 59 of the marker pairs showed *r*^2^ ≥  0.1 (Appendix A). Of these pairs, 15 pairwise comparisons were intragenic LD with an average *r*^2^ of 0.18. The seven highest *r*^2^ (0.30 < *r*^2^ < 0.64) with an average of 0.46 represented intragenic LD between ILP markers of four genes: *LIM*, *CesA3*, *APL*, and *MYB*. Forty-four marker pairs were intergenic LD, with an average *r*^2^ of 0.14. Among intergenic LD, 35 marker pairs were LD between ILP developed from transcription factors and functional genes and nine were LD between ILP developed from functional genes.

### 2.3. Association Mapping of ILP Markers

The results of phenotype statistics of GiD and GiW were described in the previous work [1]. For GiC, the mean was 4.49 cm, with the maximum of 6.96 cm and the minimum of 2.30 cm. Significant positive phenotypic correlations (*p* < 0.01) were found between GiD and GiC, and between GiW and GiC, with *r* values of 0.51 and 0.61, respectively. Different association mapping models were compared, and the proportion of significant results was calculated. Q and PCA yielded similar Q-Q plots (Appendix A), only the three models GLM, GLM + Q, and MLM + K + Q are presented for association analysis. The naive GLM model detected the largest number (25) of significant associations for nine candidate genes and three traits studied (Appendix A). This model did not take account of possible confounding effects and the associated false positives so that the cumulative distribution of *p* values was greatest from the observed *p* values. The GLM + Q models performed better than the naive model but still deviated from the expected values. These models yielded 13 significant associations for six candidate genes and all traits. The MLM + Q + K model showed the smallest departure from expectations in the Q-Q plots. It detected nine marker–trait associations for four candidate genes and three traits. All three models displayed similar distributions of *p* values for GiC, but the MLM + Q + K model still deviated less from the expected values than the GLM or GLM + Q models and detected a single marker–trait association for GiC. As the MLM + Q + K model was superior at accounting for spurious associations resulting from population structure and/or familial relatedness, the results from this model are discussed next (Table 1). A total of nine marker–trait associations involving four ILP markers developed from three candidate genes were detected. Marker–trait associations of COBL064_1, and PIN227_1 were significantly associated with GiD and GiW, using a Bonferroni corrected cut-off (with *p* ≤ 4.3 × 10^−4^). These marker–trait associations explained the phenotypic variation, with ranges from 4.9% (CaS108) to 14.6% (COBL064_1)

### 2.4. Validation of the Associated Markers in Rubber Tree Breeding Lines

For the breeding lines used to validate the markers identified by association mapping, girth was lower in the dry season than in the wet season. The mean GiD was 22.37 cm, with the largest girth of 28.66 cm and the smallest girth of 13.33 cm. The mean GiW was 26.15 cm, with a maximum of 33.28 cm and a minimum of 15.73 cm. The mean GiC was 13.49 cm, with the maximum of 21.62 cm and the minimum of 7.40 cm. Significant positive phenotypic correlations (*p* < 0.01) were found between GiD and GiC, and between GiW and GiC, with *r* values of 0.63 and 0.54, respectively.

ILP markers identified in the Amazonian population as associated with the girth growth traits were tested by looking for significant markers in the rubber tree breeding lines, to determine whether the association between marker and trait was maintained over the breeding lines. The results revealed that one allele of COBL064_1 and three of PIN227_1 remained significantly (*p* < 0.05 or 0.01) associated with GiD and GiW in the rubber tree breeding lines, and one allele of CAD76 with GiC (Table 2). In the case of COBL064_1, individuals carrying the 380-bp allele showed a significant (*p* < 0.05) reduction in GiD and GiW. In the case of PIN227_1, individuals carrying the 261-bp allele showed a significant increase in GiD and GiW, whereas individuals carrying the 289- and 298-bp alleles showed a significant reduction in both. In the case of CAD076, individuals carrying the 263-bp allele showed a significant reduction in GiC.

### 2.5. Identification of Polymorphism and Linkage Disequilibrium within COBL Gene

Given the strongest association between the ILP marker in the *COBL* gene with GID and GIW, we further investigated this particular variant using SMRT CCS technology. Three specific primer pairs yielded high-quality sequences. The sequencing of the *COBL* gene with a total length of 4236 bp revealed 128 SNPs and 178 indels. The amplified regions covered 6 exons, 5 introns, and the 3’ UTR. Of 306 polymorphic sites, 72 and 83 were located in exons, and introns, respectively, and 151 polymorphic sites within the 3’ UTR. The LD between pairs of polymorphic sites ranged from 6.7 × 10^−7^ to 1.0 with an average *r*^2^ = 0.24. Plotting *r*^2^ values against physical distance between polymorphic pairs (Figure 1) indicated the LD decay of 650 bp. i.e., the maximum physical distance for genetically linked markers. This result suggested that the LD of the SNPs within this gene did not extend over the entire gene region.

### 2.6. Association between SNP/Indel within COBL and Growth Girth

When pairs of adjacent loci were found to demonstrate LD, only one of the two SNPs from each pair was used in marker–trait association analysis, leaving 55 SNPs and 162 Indels in the candidate gene. Of these, 63, 55 and 99 polymorphic sites were located in exons, introns, and the 3’ UTR, respectively, of this gene. The FarmCPU model showed a straight line close to 1:1 with a slightly deviated tail in the Q-Q plots, suggesting that this model controlled both false positives and false negatives (Figure 2). FarmCPU showed a trend of association with GID and GIW toward 3’ UTR of the *COBL* gene. The most associated position with GID (Table 3 and Figure 3) was the SNP at 5589 bp (*p* = 6.50 × 10^−5^; *q* = 0.014) explaining 11.1% of the phenotypic variance. The suggestive association (*p* < 0.01) with GID and GIW was the SNP at 3221 in intron5.

## 3. Discussion

Increasing the girth growth is one important step to further improve economic values of the rubber tree. In the present study, we demonstrated for the first time an association between girth growth traits and allelic variations at various candidate gene loci using the ILP markers. Comparison of the EST sequences with the whole-genome shotgun contigs of the rubber tree enabled the design of 235 primer pairs for ILP markers. This primer design strategy explicitly focused on single-copy genes due to complexities generated by the presence of paralogous copies [23,24].

The high PCR success rate (97%) confirmed that the predicted exon/intron junction sites could be utilized to develop ILP markers in this study. A total of 115 polymorphic ILPs from 26 candidate genes involved in secondary growth, were selected. The PIC varied from 0.06 to 0.88, with an average of 0.44. The results were similar to the findings from EST-SSR markers [25] but higher than the results from ILP markers developed from a previous study [26]. Based on these ILP markers, model-based Bayesian analysis and PCA suggested that the accessions of Amazonian rubber trees fell into two clusters. These results were consistent with previous reports of two distinct clusters within this population, using SNPs [1] and comparable with those reported for the IRRDB collection [27,28,29]. Our results suggested that the intronic regions were sufficiently varied to act as marker resources when determining the population structure in rubber tree.

Gene-based markers are a type of marker with the most potential to bridge the gap between structural polymorphisms and functional diversity, since these gene-based markers are related to phenotypic variations. Here, we demonstrate the utility of ILP markers in identifying the best candidate gene for girth growth. COBL064_1 derived from *COBL4* showed the strongest association with girth across two seasons and explained the relatively high portion of the phenotypic variance. This marker also remained significantly associated with GiD and GiW in the rubber tree breeding lines. The *COBL4* gene is a member of the COBRA family, which encodes glycosyl phosphatidylinositol (GPI)-anchored proteins that regulate cellulose deposition and oriented cell expansion in the plant cell wall [30]. Linkage analysis in a full-sib family revealed that *COBL4* was the associated gene in a quantitative trait locus (QTL) region for cellulose content in *Eucalyptus* [10,31]. In the present study, the associated polymorphism of the COBL064_1 marker is located in the intron 5. The fact that sequence variants located within a noncoding region are associated with the variation in girth growth traits could be explained by two hypotheses. Firstly, the detected variants are in LD with causal polymorphisms in regulatory sequences that are responsible for the trait variance. Secondly, the nucleotide change within the intron is involved in regulatory functions like alternative splicing, and therefore, can affect the structure, function, and expression level of proteins [32,33]. The important next step to understanding the genetic contributions to the girth growth traits is identifying which SNP(s) are likely to be causal at this candidate gene.

In order to analyze whether genetic variants in *COBL4* might be mediating gene-based marker–trait association, we applied candidate gene association mapping to identify SNPs with the strongest association with the traits. Several studies using this approach in plants have been able to successfully identify associations between SNPs in candidate genes and phenotypic traits [3,7,34]. In the present study, the *COBL4* gene was selected for candidate gene association mapping due to the known functional relevance to growth properties, and its association with the girth growth traits in the natural population and the breeding lines. We applied targeted sequencing using SMRT technology to assess sequence variation. The long-read sequencing technologies such as PacBio SMRT can allow for the assembly of multigene family clusters and serve to more accurately characterize patterns of gene copy variation in gene families [35]. The CCS technology derives a consensus sequence from multiple passes of a single template molecule, producing accurate reads from noisy individual subreads [36,37,38].

The power of identifying SNP and mapping resolution for complex traits depend on the LD exploited in the population by the statistical model [39]. The fast LD decay over physical distance was observed in the *COBL4* in the Amazonian population.

From our results, FarmCPU, the multilocus model, provides a robust model for association mapping, effectively suppressing false positives. This result corresponds to the previous study demonstrating that the FarmCPU model could reduce both false positives and false negatives in soybean and maize with varying LD decays rates [40].

SMRT CCS of the candidate gene identified a SNP (CA5589) in the 3’ UTR most associated with the girth growth traits. The detected SNP is in LD with COBL064_1 (*p* = 0.006). The intragenic LD decayed within 650 bp in this gene. Thus, the SNP showing genetic association is likely to be the causal variant or located in close proximity to the causative polymorphisms. The 3’ UTR of messenger RNAs may contain cis-regulatory elements that affect gene expression by altering mRNA stability and translation [41]. Additional studies are needed to better elucidate the mechanisms underlying these putative associations. Overall, this study showed that the cost-effective method of ILP gene-based markers can assist in identification of SNPs in the candidate gene associated with girth growth. A combination of the SNP markers identified in this study and our previous report [1] can be applied to rubber tree breeding programs for improvement of girth growth by marker-assisted selection.

## 4. Materials and Methods

### 4.1. Plant Materials and Field Data

The Amazonian population comprised 170 accessions from three Brazilian states divided as follows: 14 from Acre, 91 from Rondoma, 64 from Mato Grosso and one of unknown origin. The details of the materials and field experiments were described in the previous study of Chanroj et al. [1]. Briefly, all trees were propagated by grafting with five replicates, and planted at a spacing of 3 × 7 m. Measurements were carried out individually on each tree. The experiment was conducted in 1994 at the Nong Khai Rubber Research Center (NKRR) of Nong Khai Province, the northeast region of Thailand. To evaluate the growth of individual trees, the girth (circumference at 1.7 m) was measured in centimeters (cm). Means were then calculated based on the total replicates. Measurements were taken in March (dry) and September (wet) in 2000, 2002, and 2004. The trees generally reached a latex tappable size. The total change in girth (GiC) was also calculated by subtraction from the circumferences in the 2000 and 2004 data. Three traits were used for association analysis, girth in the dry (GiD), and wet (GiW) seasons and GiC.

### 4.2. Candidate Gene Selection

Gene selection (Table 4) was based on earlier reports of secondary growth development and differential gene expression in *Eucalyptus* spp. [11,13,42,43] and of the genes involved in xylem/wood development in loblolly pine (*Pinus taeda* L.) [9]. Twenty six genes were selected from four function classes: four from cellulose synthesis (*CesA2, CesA3, CaS* and *COBL4*), eight from lignin synthesis (*4CL, C4H, CAD, CCoAMT, CCR, COMT, PAL,* and *Peroxidase2*, 12 associated with the transcription factors regulating secondary cell wall development (*BTF3, FRA2, HD-zip, KNAT, LIM, MOR1, MYB, NAC1, NtLIM1, PIN1, RIC1* and *APL*), and two from cell expansion (*COB,* and *KORRIGAN*).

### 4.3. ILP Marker Development

Gene transcripts were used as queries in a BLASTN program search of the whole-genome shotgun sequences of rubber tree cultivar RRIM 600 [44] in the NCBI database. Regions of similarity (E-value ≤ 10^−4^ and an identity > 65%) between selected sequences and whole-genome shotgun contigs were identified using BLAST. The position and length of introns within the rubber tree transcripts were predicted from the alignment results. The specific primers flanking the predicted intron positions were designed using the Primer3 program [45] with the default settings. Amplification by PCR was performed in a 20 µL reaction mixture containing 10 ng template DNA, 1 × PCR buffer (20 mM Tris pH 9.0, 100 mM KCl, 3.0 mM MgCl_2_), 200 µM of each of the four dNTPs, 1.5 µM of each of the forward and reverse primers, and 0.5 U Taq DNA polymerase. Thermal cycling conditions were as follows: 3 min initial denaturation at 95 °C, 30 cycles of 30 s for denaturation at 95 °C, 30 s of annealing at 52–60 °C (depending on the combination of primers), a 30 s extension at 72 °C, and a final 5 min extension at 72 °C. PCR products were separated on 6% denaturing polyacrylamide gels and silver-stained. Since large intron fragments can produce inaccurate scores when differences in fragment size are small, and can reduce the rate of PCR amplification, the polymorphic ILP markers were selected between 150 and 800 bp in length. Allelic variation was calculated from the genotypic profiles of the ILP markers based on the number of alleles and the polymorphism information content (PIC; [46]), using PowerMarker 3.25 [47]. The genetic diversity parameters included the number of alleles (Na), expected heterozygosity (He), and observed heterozygosity (Ho).

### 4.4. Population Structure and Linkage Disequilibrium Analysis of ILP Markers

To test for the potential applications, the polymorphic ILP markers were used to evaluate the genetic population structure of the Amazonian accessions used in the present study. Two methods were used to determine the population structure. First, a two-dimensional diagram of principal coordinates analysis (PCA) was produced based on a genetic distance (GD) matrix by NTSYS-pc v. 2.0 [48]. The second method used the model-based Bayesian analysis in the software STRUCTURE v. 2.3.4 [49]. The analysis was run 10 times for each K value from K = 1 to K = 10. The burn-in length was set to 100,000 and the number of iterations to 100,000. The ad hoc statistics defined by Evanno et al. [50] as ΔK were used. Based on the posterior probability of membership (Q) of a given accession, the accession was classified as an admixture with a membership probability of Q < 0.70. The squared correlation coefficients (*r*^2^) between ILP markers were used to quantify intragenic and intergenic LD, computed using TASSEL 2.1 [51]. The significance of this measure was assessed using *p*-values after Bonferroni correction for multiple tests (α = 0.05).

### 4.5. Association Analysis of ILP Markers

To assess the effect of population structure (Q) and relative kinship (K) on traits in the Amazonian germplasm accessions, Q and K were inferred from an independent set of 1820 SNPs as previously described [1] to avoid dependency among terms in the model and to prevent the structure from absorbing the QTL effects from the model [52,53]. ILP markers were used to perform association analyses through general linear models (GLM) and the mixed linear model (MLM). The TASSEL V2.1 [51] software package was used to analyze the multiallelic genotypic data of ILP markers and the phenotypic data of girth traits using the GLM without correction, the GLM model corrected with the Q-matrices (GLM + Q) or PCA (GLM + PCA), and the MLM incorporated both K and Q matrices (MLM + K + Q) or PCA (MLM + K + PCA). All models were assessed for their ability to control for type I errors by plotting the Quantile-Quantile (Q-Q) *p* values for the markers, where uniformly distributed *p* values indicate proper control. The best-fit model was selected for association analysis. For multiple comparison adjustment, the marker–trait association was considered significant if the adjusted *p*-value following FDR (*q* value) was <0.05.

### 4.6. Validation of the Identified ILP Markers

To confirm the effectiveness of the ILPs detected, markers identified by association mapping were further validated using 95 rubber tree breeding lines with a range of growth girths. These breeding lines were generated from 23 parents, propagated by grafting with three replicates, and planted in seven-tree row plots at a spacing of 3 × 7 m. The experiment was conducted in the year 2000 at NKRR. Girth measurements were taken collected continuously in March (GiD) and September (GiW) between 2003 and 2006. GiC was calculated by subtraction from the circumferences of years 2003 and 2006. These evaluation periods were treated as independent environments. DNA isolation and PCR amplification were performed as described previously for ILP marker development. The amplified product was resolved using a ZAG™ DNA Analyser.

### 4.7. Library Preparation and SMRT Sequencing of the Candidate Gene

The candidate gene, *COBL*, that encodes glycosyl phosphatidylinositol (GPI)-anchored proteins regulating cellulose deposition and oriented cell expansion in the plant cell wall, was selected for sequencing in the Amazonian population. We applied targeted sequencing using Pacific Biosciences (PacBio) single-molecule real-time (SMRT) circular consensus sequencing (CCS) to assess sequence variation. Two-step barcoding for each individual was performed with M13-tailed primers. The target regions were first amplified with pairs of gene-specific primers with M13 forward and reverse sequence tails (Appendix A). The *COBL*-specific primers were used to generate a 4236 kb fragment covering 6 exons, 5 introns, and 3’ UTR (GenBank: AJJZ010406963.1). Thermal cycling conditions were as follows: 3 min initial denaturation at 95 °C, 35 cycles of 30 s for denaturation at 95 °C, 60 s of annealing at 52–60 °C (depending on the combination of primers), a 45 s extension at 72 °C, and a final 5 min extension at 72 °C. Sample-specific barcodes were introduced in a second PCR with identical conditions as the first, but using 3 μL of the purified product from the first step PCR. A SMRT bell library and sequencing were performed as recommended by Pacific Biosciences.

### 4.8. Linkage Disequilibrium and Candidate Gene Association Mapping of COBL4

The raw reads generated from SMRT sequencing were processed using the Pacific Biosciences (CA, USA), SMRT analysis (v2.3.0) software package. SNPs within the candidate gene sequence were selected for the linkage disequilibrium analysis based on a minor allele frequency threshold of 5%. The extent of LD was estimated by nonlinear regression analysis on the basis of intragenic *r*^2^ values [1]. When pairs of adjacent loci were found to demonstrate LD, one of the two SNP/Indels from each pair was used for marker–trait association analysis. Associations between the biallelic genotypic data of SNP/Indel markers and the mean girth traits were analyzed using the mixed linear model (a single locus model) incorporated both K, and Q matrices using TASSEL v. 5.2.54 [51], and by applying FarmCPU (Fixed and random model Circulating Probability Unification; a multilocus model) [54] in R. The relative distribution of −log10 *p*-values was observed for each SNP/Indel association and compared individually with the expected distribution using the Q-Q plot. The best-fit model was selected for association analysis. To control for type I errors due to multiple testing, the marker–trait association was considered significant if the adjusted *p*-value following FDR was <0.05.

## Figures and Tables

**Figure 1 plants-10-01440-f001:**
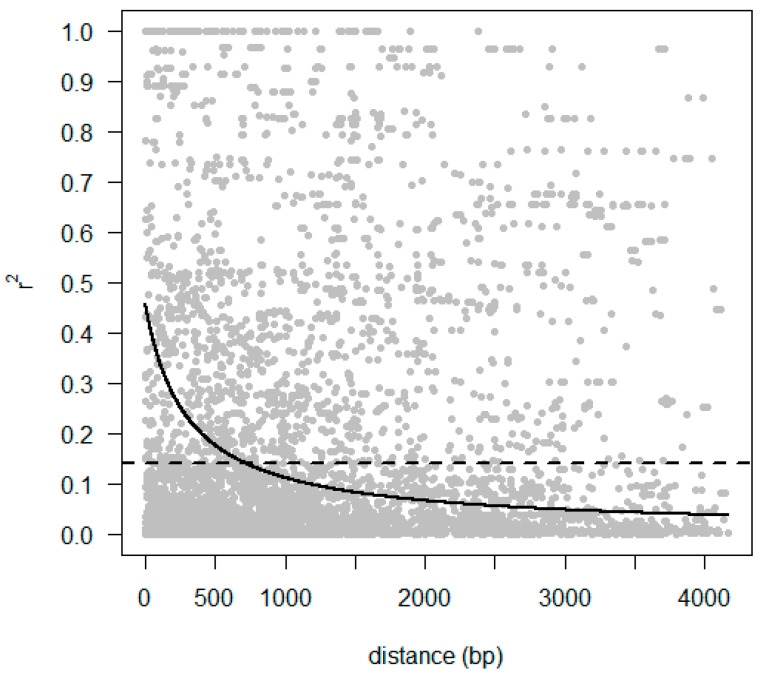
Pairwise LD (*r*^2^) values were plotted against the physical distance between all pairs of polymorphic sties (SNP/Indel) within the *COBL* gene. The trend line of the nonlinear regressions against physical distance is given by the solid line. The horizontal dotted line indicates the 95% percentile of the distribution of the unlinked *r*^2^, which gives the critical value of *r*^2^. The intersection point with the nonlinear regression curve, which determines LD decay, marks the threshold for the maximum physical distance between genetically linked markers.

**Figure 2 plants-10-01440-f002:**
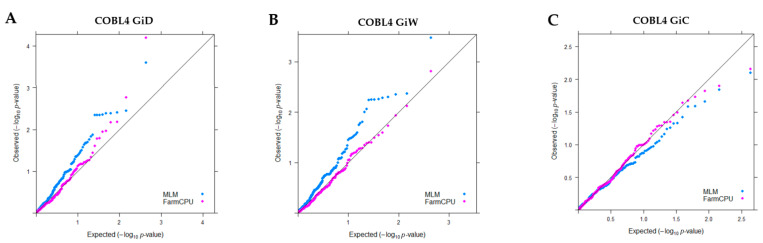
Quantile-quantile plots of estimated −log10 (*p*) from association analysis using MLM model and FarmCPU and three traits: (**A**)girth in the dry season (GiD) (**B**) girth in the wet season (GiW) and (**C**) total girth increment (GiC). The black line is the expected line under the null distribution. The blue line represents the observed *p* values using the MLM model, and the red line using FarmCPU.

**Figure 3 plants-10-01440-f003:**
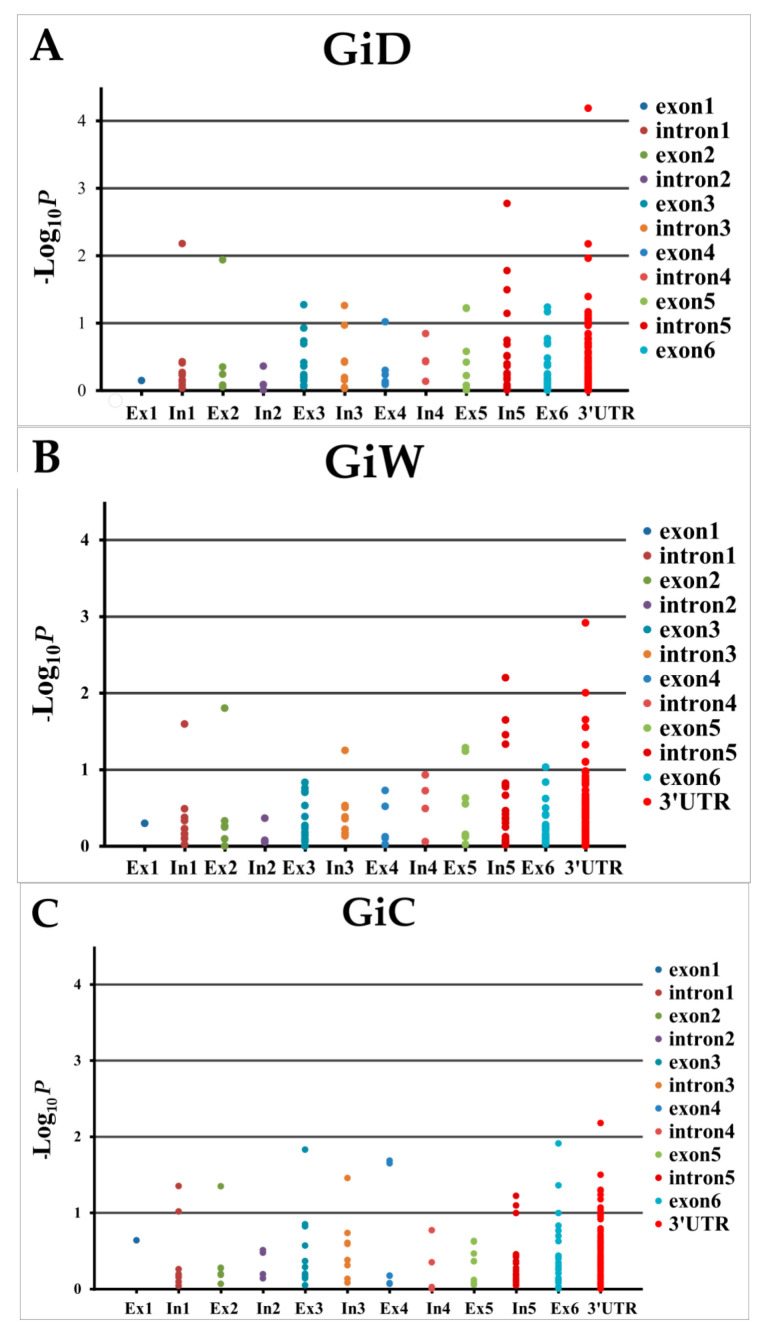
Marker–trait associations across the COBL gene. Associations were based on FarmCPU. The p-value of the marker is given at each position (*p*-value was decimal log transformed). (**A**) girth in the dry season (GiD), (**B**) girth in the wet season (GiW), and (**C**) total girth increment (GiC).

**Table 1 plants-10-01440-t001:** Markers and trait association from the MLM + Q + K model.

Trait ^1^	Marker ^2^	*p* Value	Phenotypic Variation (%)
GiD	COBL064_1 *	2.0 × 10^−7^	14.0
	PIN227_1 *	1.0 × 10^−5^	10.2
	PIN220 *	1.1 × 10^−4^	7.7
	CaS108	9.2 × 10^−4^	4.9
GiW	COBL064_1 *	4.32 × 10^−7^	14.6
	PIN227_1 *	4.82 × 10^−5^	10.0
	PIN220	4.7 × 10^−4^	7.3
	CaS108	1.8 × 10^−3^	4.9
GiC	CAD076	1.6 × 10^−3^	8.9

^1^ GiD = girth in the dry season, GiW = girth in the wet season, and GiC = total girth increment; ^2^ * Significant after Bonferroni correction *p* value = 4.3 × 10^−4^.

**Table 2 plants-10-01440-t002:** Allelic effect on girth growth traits of each significant ILP marker in rubber tree breeding lines.

Marker	Size (bp)	GiD	GiW	GiC
*p*-Value	Effect	*p*-Value	Effect	*p*-Value	Effect
COBL4	380	0.033	−1.52	0.044	−1.68	ns	−0.08
PIN227	261	0.01	1.22	0.026	1.16	ns	−0.20
289	0.005	−1.66	0.01	−1.74	ns	−0.29
298	0.013	−2.02	0.027	−2.17	ns	−0.41
PIN220	461	0.025	3.98	0.023	4.70	ns	3.32
CAD76	263	ns	−0.93	ns	−1.07	0.017	−1.49
312	ns	0.41	ns	0.49	0.043	1.25

**Table 3 plants-10-01440-t003:** Markers and trait association mapping from FarmCPU model.

Trait ^1^	Marker	Position ^2^	*p*-Value ^3^	Phenotypic Variation (%)
GiD	3’ UTR	5589	6.50 × 10^−5^ *	11.1
	intron5	3221	1.69 × 10^−3^	7.0
	intron1	1475	6.63 × 10^−3^	6.1
	3’ UTR	4406	6.69 × 10^−3^	10.1
GiW	3’ UTR	5589	1.56 × 10^−3^	9.8
	intron5	3221	7.64 × 10^−3^	6.4
GiC	3’ UTR	5526	6.89 × 10^−3^	3.6

^1^ GiD = girth in the dry season, GiW = girth in the wet season, and GiC = total girth increment; ^2^ Position in GenBank accession AJJZ010406963.1; ^3^ * Significant after FDR correction.

**Table 4 plants-10-01440-t004:** Candidate genes for secondary growth development used in the study.

Functional Class		Gene	Full Name	No. of ESTs	References
Cellulose	1	*CesA2*	Cellulose synthase2	1	[9,13]
synthesis	2	*CesA3*	Cellulose synthase3	1	[9,13]
	3	*COBL4*	COBRA-like protein 4	1	[10,13]
	4	*CaS*	Callose synthase 10	3	[9]
Lignin synthesis	5	*4CL*	4-coumarate:CoA ligase	1	[9,11,12,13]
	6	*CAD*	Cinnamyl alcohol dehydrogenase	3	[9,10,11,12,13]
	7	*COMT1*	Caffeic O-methylransferase1	2	[9,11,13]
	8	*CCoAMT*	Caffeoyl-CoA O-methyltransferase	3	[9,10,13]
	9	*CCR*	Cinnamoyl-CoA reductase	2	[9,10,11,12]
	10	*C4H1*	Cinnamate 4-hydroxylase 1	2	[10,11]
	11	*PAL*	Phenylalanine ammonia-lyase	1	[9,11]
	12	*Peroxidase2*	Peroxidase 2	1	[9]
Transcription	13	*LIM*	LIM gene for LIM transcription factor	2	[9,13]
factors	14	*MYB1*	MYB1 gene for MYB transcription factor 1	5	[9,10,13]
	15	*BTF3*	Transcription factor BTF3 homolog 4	2	[13]
	16	*HD-zip*	Homeobox-leucine zipper protein ANTHOCYANINLESS 2	1	[13]
	17	*NAC1*	NAC domain-containing protein 100-like	4	[13]
	18	*APL*	myb family transcription factor APL	1	[9]
	19	*KNAT*	Homeobox protein knotted-1-like	1	[9]
	20	*NtLIM1*	Eglim1 gene for transcription factor lim1	2	[10]
	21	*MOR1*	Protein MOR1	1	[9]
	22	*PIN1*	Auxin efflux carrier component 1-like	1	[9]
	23	*RIC1*	CRIB domain-containing protein RIC7	1	[9]
	24	*FRA2*	Katanin p60 ATPase-containing subunit A1	1	[9]
Cell expansion	25	*COB*	COBRA-like extracellular glycosyl-phosphatidyl inositol-anchored protein family	1	[9]
	26	*KORRIGAN*	Korrigan	1	[9,10]
Total	26			45	

## Data Availability

All relevant data can be found within the manuscript and Appendix A.

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
