# Peer review of "Identification of Candidate Gene-Based Markers for Girth Growth in Rubber Trees"

_plants, 2021, doi:10.3390/plants10071440_

Round 1
Reviewer 1 Report
The article concerns the identification of candidate genes related to girth growth in rubber trees. The studied trait is of high economical importance. Genetic studies on trees are always difficult due to the long period of life cycle of studied species. I find the research done by authors a very interesting and worth the publication in Plants journal.
The number of studied genotypes (170) is relatively low for association mapping, but it can be acceptable due to the specificity of studied specie. Nevertheless, the description of Plants materials and field data (section 4.1) is not fully sufficient. Authors indicated that the data were recorded from 2000 to 2014, but description concerning the character of these data are given only for the period 2000-2004. Latter, in section 4.6 (Validation…) in line 354 we can read: “The experiment was conducted in the year 2000 at NKRR” – I suppose that it was started at this year, but conducted for several following years (?). Does the measurements were collected only on 2003 and 2006 (data used for analyses), or were collected continuously through the whole period as is indicated in the text?
In my opinion, the manuscript can be published in the journal Plants after minor revision.
Author Response
Dear Reviewer 1,
We appreciate for the thoughtful comments. All comments have been responded to point-by-point. The manuscript was edited following the reviewers’ suggestions using Track Changes.
Best Regards,
Kittipat Uoskit, Ph.D
Corresponding author
---------
Reviewer 1
Comments and Suggestions for Authors
The article concerns the identification of candidate genes related to girth growth in rubber trees. The studied trait is of high economical importance. Genetic studies on trees are always difficult due to the long period of life cycle of studied species. I find the research done by authors a very interesting and worth the publication in Plants journal.
The number of studied genotypes (170) is relatively low for association mapping, but it can be acceptable due to the specificity of studied specie. Nevertheless, the description of Plants materials and field data (section 4.1) is not fully sufficient. Authors indicated that the data were recorded from 2000 to 2014, but description concerning the character of these data are given only for the period 2000-2004.
Response to the reviewer:
Field planting of the Amazonian population used in the present study was carried out in 1994. The data were recorded in the period 2000–2014, we used the data in the early tapping phase taken in March (dry) and September (wet) in 2000, 2002, and 2004. The trees generally reach a latex tappable size. For clarity, we have revised the sentences marked up using the Track Changes in the revised manuscript (section 4.1).
Latter, in section 4.6 (Validation…) in line 354 we can read: “The experiment was conducted in the year 2000 at NKRR” – I suppose that it was started at this year, but conducted for several following years (?). Does the measurements were collected only on 2003 and 2006 (data used for analyses), or were collected continuously through the whole period as is indicated in the text?
Response to the reviewer:
Yes, the measurements were collected continuously through the whole period as is indicated in the text. We have revised the sentences marked up using the Track Changes in the revised manuscript (section 4.6).

Reviewer 2 Report
The submitted paper describes the identification of gene-based markers for girth growth in rubber trees.
- 26 candidate genes were already selected based on four functional classes, and 45 EST were aligned to extract the variations.
- Please explain more about the 45 ESTs, where are they from? NCBI deposit? Why was the alignment of 45 ESTs necessary? The author mentioned the reference genome of the rubber tree as follow;
- " Gene transcripts were used as queries in a BLASTN program search of the whole genome shotgun sequences of rubber tree cultivar RRIM 600 [4] in the NCBI database. "
- However, reference [4] shows
- 4. Alomari D.Z., Eggert K., von Wiren N., Alqudah A.M., Polley A., Plieske J., et al. Identifying Candidate Genes for Enhancing Grain Zn Concentration in Wheat. Front Plant Sci. 2018, 9, 1313.
- I found the RRIM600 assembly from NCBI; and also found it do not carry gene predictions. But, in the case of genome assembly of Cultivar, GT1; https://www.ncbi.nlm.nih.gov/assembly/GCA_010458925.1, it provides gene prediction.
- Is there a reason for authors to use RRIM600 reference genome?
- Please explain more about FarmCPU and why the model should be used for the association.
- Ln. 193. FarmPCU would be a typo.
- The candidate gene approach is generally suffered by the lack of population structure information. The authors implemented population structure analysis using ILP markers that are originated from only 26 genes. Please explain more about how the ILP markers can represent the population structure. From my understanding, the population structure should be estimated by whole-genome analysis or at least well-distributed genome-wide markers.
Author Response
Dear Reviewer 2,
We appreciate for the thoughtful comments. All comments have been responded to point-by-point. The manuscript was edited following the reviewers’ suggestions using Track Changes.
Best Regards,
Kittipat Uoskit, Ph.D
Corresponding author
---------
Comments and Suggestions for Authors
The submitted paper describes the identification of gene-based markers for girth growth in rubber trees. 26 candidate genes were already selected based on four functional classes, and 45 EST were aligned to extract the variations. Please explain more about the 45 ESTs, where are they from? NCBI deposit? Why was the alignment of 45 ESTs necessary?
Response to the reviewer:
The position and length of introns within the rubber tree transcripts were predicted from the alignment results using ESTs as queries in a BLASTN program search of the whole-genome shotgun sequences of rubber tree cultivar RRIM 600. More than one ESTs may be found for one candidate gene. To explain more about the 45 ESTs, we have provided the supplementary data for 45 ESTs as shown in the revised Supplementary Table 1.
The author mentioned the reference genome of the rubber tree as follow; " Gene transcripts were used as queries in a BLASTN program search of the whole genome shotgun sequences of rubber tree cultivar RRIM 600 [4] in the NCBI database. " However, reference [4] shows
- Alomari D.Z., Eggert K., von Wiren N., Alqudah A.M., Polley A., Plieske J., et al. Identifying Candidate Genes for Enhancing Grain Zn Concentration in Wheat. Front Plant Sci. 2018, 9, 1313.
Response to the reviewer:
We have corrected the reference.
I found the RRIM600 assembly from NCBI; and also found it do not carry gene predictions. But, in the case of genome assembly of Cultivar, GT1; https://www.ncbi.nlm.nih.gov/assembly/GCA_010458925.1, it provides gene prediction. Is there a reason for authors to use RRIM600 reference genome?
Response to the reviewer:
Genome assembly of cultivar, GT1 was released in 2020. At the time (2019) we conducted this research, the draft genome sequence including 68,955 gene models of RRIM600 was reported. All 45 ESTs could be aligned perfectly with whole-genome shotgun contigs of RRIM600 to predict the position and number of the introns. Therefore, we believe that the ILP markers developed in this study could be the gene-based markers for girth growth in rubber trees and could be used for association mapping.
Please explain more about FarmCPU and why the model should be used for the association.
Response to the reviewer:
We have inserted a paragraph in the section 3 to explain more about FarmCPU and why the model should be used for the association.
Ln. 193. FarmPCU would be a typo.
Response to the reviewer:
We have corrected to FarmCPU
The candidate gene approach is generally suffered by the lack of population structure information. The authors implemented population structure analysis using ILP markers that are originated from only 26 genes. Please explain more about how the ILP markers can represent the population structure. From my understanding, the population structure should be estimated by whole-genome analysis or at least well-distributed genome-wide markers.
Response to the reviewer:
The polymorphic ILP markers were used only to test for the potential applications to evaluate the population structure of the Amazonian accessions used in the present study. The population structure and kinships used for association analysis were estimated by whole-genome analysis using an independent set of 1,820 SNPs as previous described [1] and already mentioned in 4.5.
